# Comparison of [^18^F]FDG PET/CT and MRI for Treatment Response Assessment in Multiple Myeloma: A Meta-Analysis

**DOI:** 10.3390/diagnostics11040706

**Published:** 2021-04-15

**Authors:** Kota Yokoyama, Junichi Tsuchiya, Ukihide Tateishi

**Affiliations:** Department of Diagnostic Radiology, Tokyo Medical and Dental University, Tokyo 113-8510, Japan; tuwu11@gmail.com (J.T.); ttisdrnm@tmd.ac.jp (U.T.)

**Keywords:** [^18^F]FDG PET, PET/CT, MRI, multiple myeloma, treatment response assessment

## Abstract

The present study was designed to assess the additional value of 2-deoxy-2[^18^F]fluoro-D-glucose ([^18^F]FDG) positron emission tomography/computed tomography (PET/CT) to magnetic resonance imaging (MRI) in the treatment response assessment of multiple myeloma (MM). We performed a meta-analysis of all available studies to compare the detectability of treatment response of [^18^F]FDG PET/CT and MRI in treated MM. We defined detecting a good therapeutic effect as positive, and residual disease as negative. We determined the sensitivities and specificities across studies, calculated the positive and negative likelihood ratios (LR), and made summary receiver operating characteristic curves (SROC) using hierarchical regression models. The pooled analysis included six studies that comprised 278 patients. The respective performance characteristics (95% confidence interval (CI)) of [^18^F]FDG PET/CT and MRI were as follows: sensitivity of 80% (56% to 94%) and 25% (19% to 31%); specificity of 58% (44% to 71%) and 83% (71% to 91%); diagnostic odds ratio (DOR) of 6.0 (3.0–12.0) and 1.7 (0.7–2.7); positive LR of 1.8 (1.3–2.4) and 1.4 (0.7–2.7); and negative LR of 0.33 (0.21–0.53) and 0.81 (0.62–1.1). In the respective SROC curves, the area under the curve was 0.77 (SE, 0.038) and 0.59 (SE, 0.079) and the Q* index was 0.71 and 0.57. Compared with MRI, [^18^F]FDG PET/CT had higher sensitivity and better DOR and SROC curves. Compared with MRI, [^18^F]FDG PET/CT had greater ability to detect the treatment assessment of MM.

## 1. Introduction

Multiple myeloma (MM) is the second most common hematologic cancer, and frequently develops in the elderly, with 65 years as the median age at diagnosis. MM accounts for approximately 1% of cancers and 10% of all hematologic malignancies [1,2]. It is characterized by the proliferation of plasma cells that produce abnormal monoclonal immunoglobulin, which infiltrates the bone marrow (BM), and by the excessive production of monoclonal immunoglobulins that can be detected in serum and/or urine [1]. The major clinical manifestations of MM have been given the acronym CRAB, which comprise hypercalcemia (C) and bone destruction (B) secondary to overactivation of osteoclasts; renal impairment (R), which is mostly caused by monoclonal light chains that affect the kidneys, and anemia (A) [1]. BM involvement and extramedullary disease (EMD) have been recognized as important factors that influence the prognosis and clinical management of patients with MM [3,4].

Because 80% to 90% of all patients with MM develop BM involvement [5], imaging has an increasingly important role in the assessment of the degree of BM involvement and treatment effect. A conventional radiographic skeletal survey was historically used for the assessment of bone lesions in patients with MM. However, many studies have shown higher sensitivity for the detection of focal lesions with whole-body computed tomography (CT), 2-deoxy-2[^18^F]fluoro-D-glucose ([^18^F]FDG) positron emission tomography/computed tomography (PET/CT) and whole-body magnetic resonance imaging (MRI) than with conventional radiographic skeletal surveys. Among these, whole-body MRI is the most sensitive technique for the visualization of focal and diffuse infiltration of the BM in untreated patients [6,7]. Similarly, [^18^F]FDG PET/CT has been recognized as a sensitive tool to assess the extent of both BM involvement and EMD in newly diagnosed MM [8,9,10]. Both MRI and [^18^F]FDG PET/CT upon diagnosis and during follow-up have been shown to be of prognostic value for progression-free survival (PFS) and/or overall survival (OS) [7,11].

Studies that compared the diagnostic performance between whole-body MRI and [^18^F]FDG PET/CT for the initial pretreatment staging of MM suggested that MRI had higher sensitivity in lesion detection, allowing the detection of both diffuse BM infiltration and focal lesions before destruction of the mineral bone [6,12,13,14]. Although, the International Myeloma Working Group (IMWG) recommended whole-body CT or PET/CT for first-line imaging to detect osteolytic lesions, the Myeloma Response Assessment and Diagnosis System (MY-RADS) guidelines and the National Institute for Health and Care Excellence (NICE) guidelines in the UK recommend MRI as the first-line of imaging for all patients with a suspected diagnosis of asymptomatic myeloma or solitary bone plasmacytoma [5,15,16]. The role of [18F]FDG PET/CT remains limited in the initial assessment. However, in treatment assessment, [^18^F]FDG PET/CT, compared with MRI, was associated with faster normalization of imaging findings in patients who had achieved complete response (CR) or very good partial response (PR) after therapy [7,17,18,19]. Moreover, compared with MRI, [^18^F]FDG PET/CT showed better prognostic value in detecting EMD and minimal residual disease [5,9,11,20]. Current evidence shows that [^18^F]FDG-PET is a sensitive tool for the evaluation of early therapeutic response in MM. Many studies have shown the value of [^18^F]FDG PET/CT in the treatment response assessment in MM; however, the IMWG recommendation is still weak [5]. Therefore, strong evidence on the routine use of [^18^F]FDG PET/CT in clinical decision-making is required. The present study was designed to perform a meta-analysis of all available studies and to assess the advantage of additional [^18^F]FDG PET/CT over MRI in the treatment response assessment in MM.

## 2. Materials and Methods

### 2.1. Data Sources Eligibility

We searched PubMed/MEDLINE, SCOPUS, and Biological Abstracts from inception to January 2021. Search queries included terms related to “multiple myeloma,” “[18F]FDG PET,” “MRI,” and “treatment response”. Different keywords including Medical Subject Heading (MeSH) terms were combined using Boolean operations ‘AND’ and ‘OR’ such as (“PET” OR “Positron Emission Tomography”[MeSH]) AND (“FDG”[MeSH] OR “Fluoro-Deoxy-Glucose”) AND (“MRI”[MeSH] OR “Magnetic resonance imaging” OR “DWI” OR “diffusion weighted imaging” OR “ADC”) AND (“Multiple myeloma [MeSH]” OR “MM”) AND (“response” OR “surveillance” OR “treatment effect”). We did not apply any language restrictions. Two reviewers (KY and JT) independently assessed the potentially relevant citations for inclusion, and disagreements were resolved by consensus. The referenced articles of the retrieved articles were screened to identify additional relevant articles.

The inclusion criteria were made based upon the Patient/Intervention/Comparator/Outcome/Study design (PICOS) criteria [21] as follows: (1) “patients” with MM that were given treatment, such as chemotherapy and/or stem cell transplantation (SCT), (2) [^18^F]FDG PET or PET/CT for the treatment response assessment or surveillance after treatment as the “intervention,” (3) MRI as the “comparator”, (4) detection of treatment response with IMWG Uniform Response Criteria for Multiple Myeloma as “outcomes,” and (5) the “study design” included prospective or retrospective studies published as original articles or brief communications. The exclusion criteria were as follows: (1) small sample size (<10 patients), (2) were not in the field of interest, (3) had an overlap in the study population with another report, or (4) had insufficient information for derivation of 2 × 2 tables to draw a summary receiver operating characteristic curve (SROC). Reviews, guidelines, consensus statement, editorial, letters, book chapter, case reports, and meeting abstracts were also excluded.

### 2.2. Data Extraction

We independently extracted data from the eligible studies and resolved any issues by consensus. We recorded the author names; journal; publication year; origin country; number of patients; age; inclusion and exclusion criteria; study design; treatment details; imaging details, such as imaging system (PET or PET/CT; 1.5T or 3T MRI) and protocols; number of experts who interpreted the images; and the qualitative or quantitative definition of a positive test result. The remission or relapsed status in the reference standard was recorded. In addition, the numbers of patients who showed treatment effects as good responders and poor responders on [^18^F]FDG PET/CT and MRI were extracted; 2 × 2 tables were created and included the numbers of true negative (TP), false positive (FP), false negative (FN), and true negative (TN). We defined detecting a good therapeutic effect as positive, and residual disease as negative. Data extracted from publications alone were deemed adequate for this meta-analysis without contacting the authors for more information. Two reviewers (KY and JT) individually applied the widely used Quality Assessment of Diagnostic Accuracy Studies-2 tool to assess the quality of the systematic reviews of diagnostic studies [22].

### 2.3. Statistical Analysis

To evaluate the diagnostic value of [^18^F]FDG PET/CT in the treatment response assessment of MM, we calculated the pooled estimates of sensitivity, specificity, positive likelihood ratio (PLR), negative likelihood ratio (NLR), diagnostic odds ratios (DORs), and their 95% confidence intervals (CI) for both, in [^18^F]FDG PET/CT and MRI. The calculation of LRs combined sensitivity and specificity. The PLR was defined as the ratio of sensitivity over (1-specificity), and the NLR was defined as the ratio of (1-sensitivity) over specificity. The DOR was calculated as the ratio of PLR relative to NLR, with higher values indicating better performance. In addition, SROCs were drawn, and area under the curve (AUC) and Q* index were obtained. The Q* index is the best statistical method to reflect diagnostic value and was defined by the point where the sensitivity and specificity were equal; it represented the point that was closest to the ideal top left corner of the SROC space [23]. The degree of heterogeneity among the different studies was tested using the I2 test. When significant heterogeneity (i.e., I2 value of >50%) was observed, a random effect model was applied; in the other cases, a fixed effect model was used [24]. Analyses were performed using Meta-Disc v. 1.4 (XI Cochrane Colloquium, Barcelona, Spain) and Review Manager (version 5.3; Cochrane Collaboration).

## 3. Results

### 3.1. Article Search

The initial electronic search retrieved 363 articles (Figure 1). Of these, 21 articles were retrieved with the corresponding full texts after removing duplicate and nonrelevant articles. Six original articles, including a total of 278 patients, were finally included in the quantitative synthesis [7,13,17,18,19,25]. The quality assessment of the six included studies is shown in Figure 2. A total of 278 patients with MM were analyzed for the diagnostic accuracy of treatment effect assessment of the imaging methods. All 278 patients were evaluated with MRI, and 272 of them were evaluated with additional [^18^F]FDG PET/CT.

Of the six eligible studies, four were prospective and two were retrospective. All the studies enrolled patients with MM and had undergone [^18^F]FDG PET/CT and MRI for treatment assessment. One study used 3T MRI and three studies used 1.5T MRI; two studies had no available details. In three studies, the whole-body was scanned through coronal images, whereas the other three studies acquired whole spine and pelvic images. Four studies used contrast medium and two studies used noncontrast images only. All studies used [^18^F]FDG PET/CT. The amounts of radiotracer injected during [^18^F]FDG PET/CT were 350, 360 and 370 Mbq, and 3–7 Mbq/kg and 5.3 Mbq/kg, respectively. The time interval was 60–80 minutes in all studies. The results of all studies were evaluated qualitatively. The reference standard comprised BM biopsy, laboratory data, MRI, and [^18^F]FDG PET/CT.Most of the studies followed the international uniform response criteria for MM [26,27]. The detailed characteristics of included studies are summarized in Table 1.

### 3.2. Pooled Diagnostic Performance (Meta-Analysis)

Figure 3 and Figure 4 show the combined SROC curves and their corresponding findings for MRI and [^18^F]FDG PET/CT, respectively. The findings of imaging modality are described below.

-[^18^F]FDG PET or PET/CT

This pooled analysis included six studies that comprised 273 patients. The performance characteristics (95% CI) were sensitivity, 80% (56% to 94%); specificity, 58% (44% to 71%); DOR, 6.0 (3.0–12.0); PLR, 1.8 (1.3–2.4); and NLR, 0.33 (0.21–0.53). In the SROC curves, the AUC was 0.77 (SE, 0.038), and the Q* index was 0.71.

-MRI

This pooled analysis included six studies that comprised 278 patients. The performance characteristics (95% CI) were sensitivity, 25% (19% to 31%); specificity, 83% (71% to 91%); DOR, 1.7 (0.7–4.5); PLR, 1.4 (0.7–2.7); and NLR, 0.81 (0.62–1.1). In the SROC curves, the AUC was 0.59 (SE, 0.079), and the Q* index was 0.57.

-Comparison of imaging modalities

In summary, compared with MRI, [^18^F]FDG PET/CT had higher sensitivity (80%) and DOR (6.0). The SROC curves of the [^18^F]FDG PET/CT studies showed excellent diagnostic performance (Figure 3 and Figure 4). No publication bias was observed based on the results of Funnel plot analysis (Appendix A).

For each test, we combined the summary receiver operating characteristic curves and their corresponding findings. FN, false negative; FP, false positive; TN, true negative; TP, true positive.

## 4. Discussion

Compared with MRI, [^18^F]FDG PET/CT had excellent detectability of treatment response in MM with higher sensitivity. Overall, the value of [^18^F]FDG PET/CT for the detection of early treatment response was higher compared with that of MRI. We believe that the main reason for the low sensitivity of MRI was the inability to distinguish between scar tissue and active disease after treatment. Previous studies on [^18^F]FDG PET/CT after treatment have shown a significant association between early metabolic response and histopathologic tumor regression [28,29]. In addition to its ability to detect early treatment response, [^18^F]FDG PET/CT showed a good prognostic value in MM. Completely negative [^18^F]FDG uptake after treatment was proven to correlate with a high-quality response to therapy in MM [10,11]. In two studies, [^18^F]FDG PET/CT was evaluated either on day seven after induction treatment or before the first autologous SCT. Persistence of >3 focal lesions on day seven was an early predictor of significantly shorter PFS and OS, especially in the subgroup of patients with high-risk gene expression profiles [30]. Although the value of [^18^F]FDG PET/CT in the assessment of treatment response has been well established, the recommendation for its routine use in clinical settings remains weak [5,31]. To our knowledge, this was the first meta-analysis that compared the diagnostic ability between [^18^F]FDG PET/CT and MRI in the assessment of treatment response in MM. Our results show evidence for the clinical use of [^18^F]FDG PET/CT in MM.

Based on the results of this meta-analysis, the performance of [^18^F]FDG PET/CT in the assessment of treatment effect was satisfactory, based on the high percentage of true positives and the low percentage of false negative results. On the other hand, the specificity was lower with [^18^F]FDG PET/CT than with MRI. A possible reason for this was the relatively high detection rate of transient or extremely fast response in patients who were yet to be assessed to have PR or very good PR according to the response criteria. The fact that [^18^F]FDG PET/CT showed good prognostic value might support the idea, because patients who have false positive results (i.e., persistent abnormal plasma cells on BM biopsy with positive treatment response on [^18^F]FDG PET/CT, are expected to have a high probability of good prognosis according to studies [11,19]. Therefore, patients who show good response on [^18^F]FDG PET/CT should be carefully followed-up and can be expected to achieve a good remission state. In follow-up, IMWG response criteria determine [^18^F]FDG uptake as positive or negative, depending on whether it is higher or lower than blood or surrounding normal tissue. However, in clinical practice, the diagnosis may be equivocal, as bone marrow also exhibits mild [^18^F]FDG uptake in response to normal metabolic or hematopoietic activity, and degenerative uptake may be seen in older generations. In the future, a scoring system is expected to be constructed that can evaluate more objectively in combination with CT and MRI.

In the present study, the MRI protocols mainly used conventional T1 weighted image (T1WI), T2 weighted image (T2WI), short TI inversion recovery (STIR), and contrast enhanced T1WI. However, a good diagnostic value of whole-body diffusion-weighted image (WBDWI) or PET/MRI has been reported in the treatment assessment of patients with MM [30,32,33,34,35]. The Myeloma Response Assessment and Diagnosis System (or MY-RADS) [16], a standardized evaluation method proposed in 2019, also recommends WBDWI according to its usefulness in previous reports [32,36,37,38]. These studies claim that WBDWI can detect more lesions, but it has not been confirmed whether the lesions detected are true lesions. The possibility of post-treatment changes or hyperplastic bone marrow has not been ruled out [39,40,41]. One study reported that [^18^F]FDG PET/CT was superior in detecting extramedullary lesions in treatment response assessment [42]. The fact that only intramedullary lesions are more detected in WBDWI indicates that it may also include benign changes such as hyperplastic bone marrow. In most studies, WBDWI detects more focal lesions than FDG PET/CT in newly diagnosed MM, but on a per-patient basis WBDWI and [^18^F]FDG PET-CT showed no differences in the detection of bone involvement, with almost perfect agreement between the two methods [43]. In addition, [^18^F]FDG PET-CT could be considered a high-performance imaging procedure for diagnosis of diffuse infiltration [43]. Nowadays, many studies on WBDWI have focused on the detection of BM lesions in newly diagnosed MM [40,44], and there are few studies assessed the correlation with IMWG response criteria in post-treatment evaluation [38]. A study that was not included in the quantitative evaluation of the present meta-analysis, due to the small number of cases, compared the ability of WBDWI and FDG PET/CT to evaluate treatment response in eight cases of MM. According to this, the sensitivity, specificity, and diagnostic accuracy were 60%, 33% and 50% for PET, and 20%, 67%, and 37.5% for MRI, respectively [38]. Although the number of cases was small and not statistically significant, [^18^F]FDG PET/CT appeared to be superior to WBDWI in assessing treatment response. Further validation studies that compare [^18^F]FDG PET/CT with WBDWI or PET/MRI are warranted.

The present study had certain limitations. First, the diagnostic value of [^18^F]FDG PET/CT in patients with MM was reported without adjusting for potential confounders, such as the grade and stage of MM. Moreover, this meta-analysis included a source of heterogeneity (i.e., publication bias), which may result in systematic differences in the derived effect size estimates from small and large studies. The available data from the enrolled studies consistently showed great improvement in sensitivity with [^18^F]FDG PET/CT over MRI. Although the confidence intervals included a low sensitivity for MRI and a low specificity for [^18^F]FDG PET/CT, the number of studies was limited to only those that had available data for deriving 2 × 2 tables to draw SROC. There was a risk for subjective interpretation because the [^18^F]FDG PET/CT in all the studies was qualitatively assessed by two reviewers; in addition, the presence/absence of blinding was unclear. Although the cost and radiation exposure of each procedure are important concerns for both the patient and the healthcare system, we were unable to analyze these because of the limited number of published articles. Selection of the most appropriate imaging test in any clinical situation would depend on the circumstances of the patient, expertise, available equipment at the treating site and procedure costs. Certainly, the goal is to use the most cost-effective imaging method that would allow accurate diagnosis and prompt treatment of this common, complex and costly condition. This will require further prospective studies on a larger sample size, with a direct comparison among the different radiologic and nuclear medicine techniques.

Nevertheless, our study provided evidence to support the possible application of [^18^F]FDG PET/CT, in addition to MRI, for the assessment of treatment response in patients with MM. Compared with MRI, [^18^F]FDG PET/CT had higher sensitivity for the detection of early treatment response. Further research is required to confirm the incremental diagnostic improvement with [^18^F]FDG PET/CT over MRI, with consideration of radiation dose, cost-effectiveness and the potential complications against the yield of information.

## Figures and Tables

**Figure 1 diagnostics-11-00706-f001:**
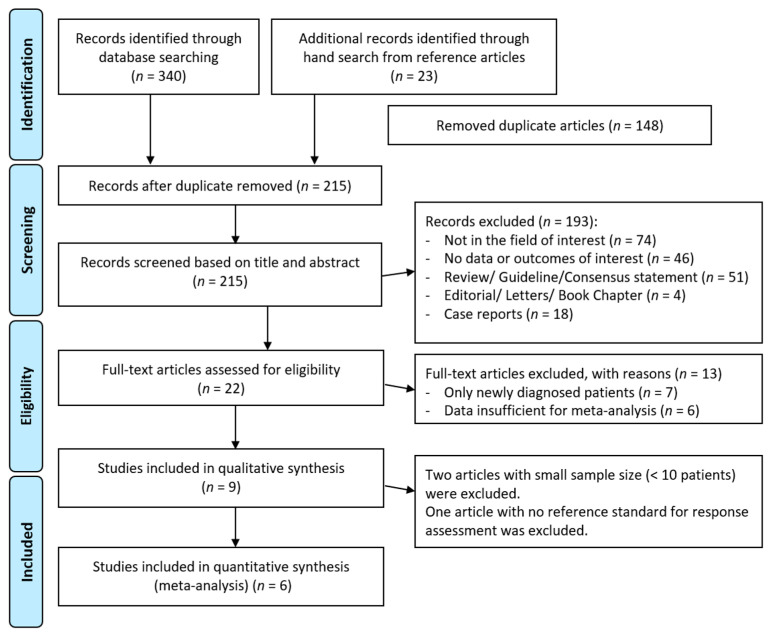
PRISMA flow diagram of included articles. PRISMA, Preferred Reporting Items for Systematic Reviews and Meta-Analyses.

**Figure 2 diagnostics-11-00706-f002:**
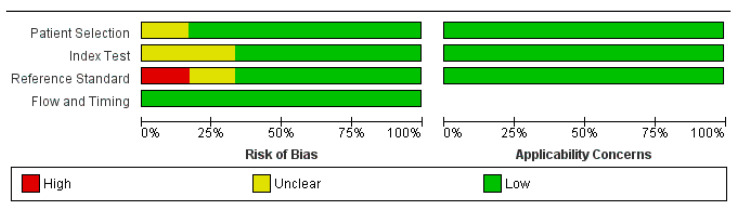
Quality assessment of the included studies using QUADAS-2.

**Figure 3 diagnostics-11-00706-f003:**
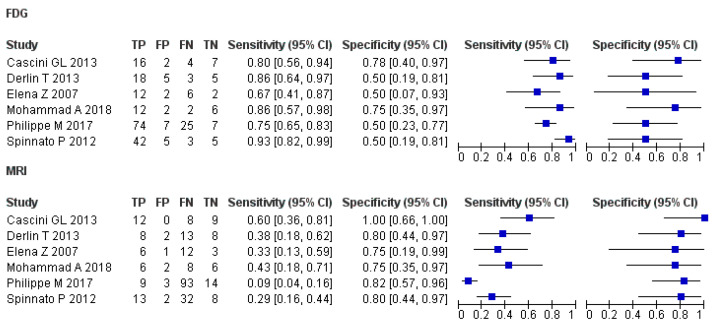
Forest plots of the sensitivity and specificity and the pooled diagnostic performance of [^18^F]FDG PET/CT and MRI.

**Figure 4 diagnostics-11-00706-f004:**
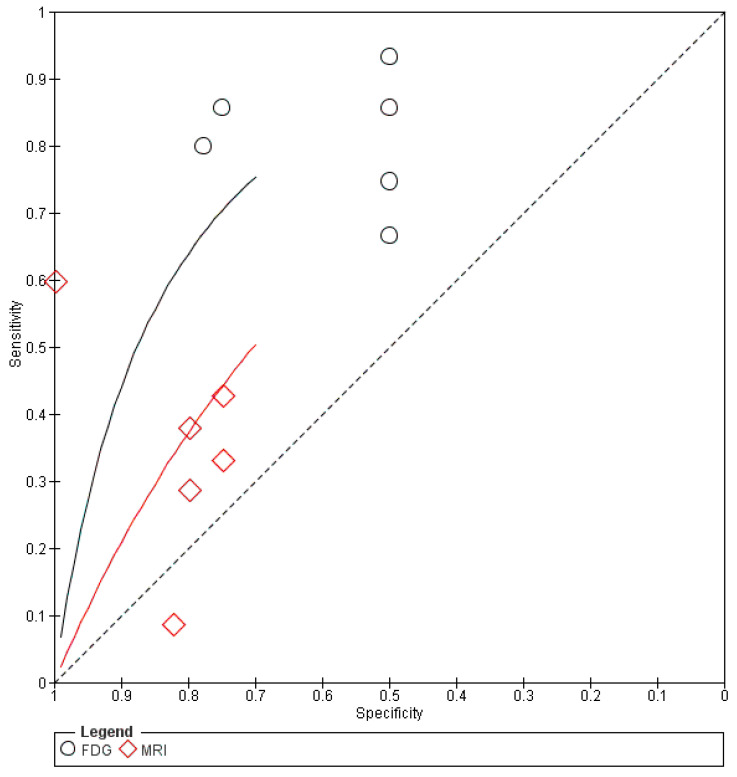
Summary receiver operating characteristic curves (SROC) for the diagnostic performance of [^18^F]FDG PET/CT and MRI. The size of the circle diamond indicates the weight of each study in [^18^F]FDG PET/CT (black curve) and MRI (red curve) respectively. The area under the SROC curve is 0.76 for [^18^F]FDG PET/CT and 0.59 for MRI.

**Table 1 diagnostics-11-00706-t001:** Study characteristics for selected studies.

Author	Year	Ref.	Country	N	Mean Age (Range)	Design	Treatment Protocol	Modality	FDG Dose	Time Interval	Fasting Time	No. of Assessors	Reference Standard	Duration
Cascini GL, et al.	2013	[25]	Italy	22	NR (44–83)	P	C or SCT	PET/CT, 1.5T MRI	360 MBq	60 min	6 h	NR	IMWG criteria [27]	2006.7–2010.3
Derlin T, et al.	2013	[7]	Germany	31	55 (38–73)	R	SCT	PET/CT, 1.5T MRI	350 MBq	60 min	4 h	2	IMWG criteria [27]	2008.7–2009.3
Elena Z, et al.	2007	[13]	Italy	22	55 (42–65)	P	SCT	PET/CT, MRI	5.3 MBq/kg	60–80 min	6 h	2	EBMT criteria [26]	2003.3–2004.12
Mohammad A, et al.	2018	[18]	Egypt	22	62 (41–78)	P	18: C, 3: Ra, 1: SCT	PET/CT, 3T MRI	370 MBq	60 min	6 h	3	BM aspiration and biopsy	2015.8–2017.12
Philippe M, et al.	2017	[19]	France	119	59 (37–65)	P	C or SCT	PET/CT, MRI	3-7 MBq/kg	60–80 min	4 h	2	IMWG criteria [27]	2010.11–2012.11
Spinnato P, et al.	2012	[17]	Italy	55	62 (33–81)	R	C or SCT	PET/CT, 1.5T MRI	NR	NR	NA	2	biopsy, L/D, MRI, PET/CT	2009–2012.3

N number of patients, NR not reported, P prospective, R retrospective, C chemotherapy, SCT stem cell transplantation, Ra radiotherapy, IMWG International Myeloma Working Group, EBMT European Group for Blood and Marrow Transplant, BM bone marrow, L/D laboratory data.

## Data Availability

Data is contained within the article or Appendix A.

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
