# Peer review of "Comparison of [^18^F]FDG PET/CT and MRI for Treatment Response Assessment in Multiple Myeloma: A Meta-Analysis"

_diagnostics, 2021, doi:10.3390/diagnostics11040706_

Round 1

Reviewer 1 Report

The authors conducted a meta-analysis which evaluate the diagnostic value of 18-FDG PET/CT and MRI for the evaluation of treatment response of Multiple Myeloma.

-Introduction : I do not agree with the authors that the IMWG recommends to perform MRI for baseline imaging. PET/CT or CT are recommended over MRI for baseline imaging of suspected multiple myeloma (Hillengass, Lancet Oncol, 2019). 

-Material and methods: I have interrogations about how the authors managed to retrieve TP,TN,FP and FN results from the study by Moreau et al (JCO, 2017) which only provide survival data based on imaging results. If the authors did not ask for supplemental data (as mentioned in their manuscript) how did they proceed? Also, why the authors did not include Usmani et al (Blood, 2013) study in their analysis?

-Although methodology remains clear, results section and the abstract can be misleading at first because here sensitivity represents the capacity of the imaging modality to detect a disease response to treatment. Indeed, in studies related to PET/CT and MRI in MM, sensitivity rather conventionally represents the capacity of the imaging modality to detect residual disease. The authors must mention more clearly in the abstract conclusion, in the methods section and in the figures captions that the diagnostic performance are for the detection of disease response.

-A word should be put in the Discussion about the low sensitivity of MRI in the authors study (i.e. the low specificity of MRI in other studies) due to its inability to differentiate scar tissue from active disease after treatment.

-Table 1 : Moreau et al : treatment protocol comprises ASCT and not Chemo only

Author Response

General Comments

We thank the reviewer for the helpful comments and suggestions, which we believe have strengthened the manuscript substantially. We have addressed all comments in our revised manuscript and discuss them in the attached file on a point-by-point basis. All changes to the revised manuscript are presented in red font in the manuscript.

Reviewer 2 Report

The paper ”Comparison of [18F]FDG PET/CT and MRI for Treatment Response Assessment in Multiple Myeloma” by Yokoyama, Tsuchiya, and Tateishi presents a comprehensive systematic review in three major databases and respective meta-analyses thereon. The statistical analysis is state of the art, and the application of QUADAS-2 is highly appreciated and informative. This is a neat nice, relevant, and informative piece of work to which the authors are congratulated. Please find below suggestions for improvements and detailed comments.

Major

L.20, L.173, L.178: from Fig. 4 and its legend, the areas under the SROC curves appear to be 0.81 for PET/CT and 0.76 for MRI. Please correct both in the abstract and in the Results section.

L.21-23 & l.271: The conclusion in the abstract suggests that PET/CT should replace MRI, whereas the conclusion at the end of the text indicates that PET/CT should supplement MRI. Please decide which one it and unify. By the way: with pooled sensitivity/specificity pairs of 80%/58% for PET/CT and 25%/83% for MRI which support a good sensitivity, but insufficient specificity for PET/CT and vice versa for MRI, it does not seem straightforward how to make use of these two imaging modalities’ results in practice. Please discuss.  

L.74-80: the search terms are sketched, but an interested reader would need more detailed information to be able to reproduce your search findings. Please provide detailed search blocks for at least PubMed/MEDLINE and Scopus as Supplemental Material, thanks.

L.132 / L.154, Fig. 2 / L.147-148 / L.260: impressively little bias was observed across the 6 studies, especially as you explicitly state the reference standard comprised the index tests (actually, both PET/CT and MRI; L.147-148) and the level of blinding was unclear (L.260); see “Reference standard” in Fig. 2 with potential bias issues in 2 out of 6 studies. The incorporation of PET/CT and/or MRI into the reference standard increases the risk of incorporation bias, see, for instance, the following two references:

Hall MK, Kea B, Wang R. Recognising Bias in Studies of Diagnostic Tests Part 1: Patient Selection. Emerg Med J. 2019 Jul;36(7):431-434. doi: 10.1136/emermed-2019-208446.

Kea B, Hall MK, Wang R. Recognising bias in studies of diagnostic tests part 2: interpreting and verifying the index test. Emerg Med J. 2019 Aug;36(8):501-505. doi: 10.1136/emermed-2019-208447.

General: apart from the I-squared statistic regarding heterogeneity between studies, funnel plots are common to illuminate potential publication bias. Please consider adding these as Supplemental Material.

General: in other cancer forms, RECIST is predominantly used and PERCIST may seem to play a more dominant role in the future (see, for instance, Hildebrandt MG et al. FDG-PET/CT for Response Monitoring in Metastatic Breast Cancer: Today, Tomorrow, and Beyond. Cancers (Basel). 2019 Aug 15;11(8):1190. doi: 10.3390/cancers11081190); is this also imaginable in MM? Please discuss if relevant, thanks.

Minor

L.9: please add / between “tomography” and “computed”

L.75: replace ‘response; Different’ by ‘response”. Different’

L.117: replace “95% CI” by “95% confidence intervals (CI)” in order to introduce the abbreviation at its first appearance in the text

L.117: replace “both in and [18F]FDG PET/CT.” by “both [18F]FDG PET/CT and MRI.” – please stick consequently to the order first PET/CT, then MRI which you apparently have obeyed otherwise consequently

L.132: replace “21” by “22” – from Fig. 1 it appears to be 22 articles here

L.151, Fig. 1: please move the box “Removed duplicate articles (N=148)” to the right hand side of the figure; then, the flow from the top (340+23) minus these 148 gives N=215 articles on the next level, and the logic is consistent that removed articles are shown to the right, thanks.

L.166: Figures 3 and 4 show (not shows)

L.169: use abbreviation MRI

L.171: 95% CI (not confidence interval; abbreviation needs to be introduced earlier, see comment regarding L.117 above)

L.169-178: reverse order by letting results for MRI and PET/CT switch place; then, PET/CT is consequently described before MRI (which is logical as PET/CT is the index test under primary investigation)

L.182: Figures 3 and 4 (not Figure)

L.215, L.235: the abbreviations VGPR and FL are not introduced in the text before

L.224: there has been (not have)  

Author Response

General Comments

We thank the reviewer for the helpful comments and suggestions, which we believe have strengthened the manuscript substantially. We would like to express our gratitude to you for carefully reviewing not only the logical structure but also small mistakes in grammar and numerical values. We are really grateful for pointing out some important mistakes. We have addressed all comments in our revised manuscript and discuss them in the attached file on a point-by-point basis. All changes to the revised manuscript are presented in red font in the manuscript.
